# EGFR Protein Expression in KRAS Wild-Type Metastatic Colorectal Cancer Is Another Negative Predictive Factor of the Cetuximab Therapy

**DOI:** 10.3390/cancers12030614

**Published:** 2020-03-06

**Authors:** Andrea Uhlyarik, Violetta Piurko, Zsuzsanna Papai, Erzsebet Raso, Erika Lahm, Edina Kiss, Marta Sikter, Jozsef Vachaja, Istvan Kenessey, Jozsef Timar

**Affiliations:** 1Department of Oncology, Medical Center, Hungarian Defence Forces, 1065 Budapest, Hungary; zspapai@gmail.com (Z.P.); erikalahm@gmail.com (E.L.); edina.kiss.dobos@gmail.com (E.K.); martasikter@gmail.com (M.S.); buksid7@gmail.com (J.V.); 22nd Department of Pathology, Semmelweis University, 1091 Budapest, Hungary; kalocsane_piurko.violetta@med.semmelweis-univ.hu (V.P.); raso.erzsebet@med.semmelweis-univ.hu (E.R.); kenessey.istvan@med.semmelweis-univ.h (I.K.); jtimar@korb2.sote.hu (J.T.)

**Keywords:** metastatic colorectal cancer, Cetuximab, EGFR protein, survival

## Abstract

The selection of colorectal cancer patients for anti-epidermal growth factor receptor (EGFR) antibody therapy is based on the determination of their RAS mutation status—a strongly negative predictive factor—since the protein target, EGFR, is not a reliable predictor of therapeutic response. In this study, we revisited the EGFR protein issue using a cohort of 90 patients with KRAS exon2 wild-type colorectal cancer who have been treated with cetuximab therapy. Twenty-nine of these patients had metastatic tissue available for analysis. The level of EGFR protein expression in the patients was determined by immunohistochemistry and evaluated by H-score (HS) methodology. Progression-free survival (PFS) and overall survival (OS) of the patients were determined according to the EGFR-HS ranges of both the primary and metastatic tissues using Kaplan–Meyer statistics. In the case of primary tumors, EGFR scores lower than HS = 200 were associated with significantly longer OS. In the case of metastatic tissues, all levels lower than the EGFR-HS range chosen were associated with significantly longer OS. These results are explained by the fact that metastatic tissues rarely maintained the expression levels of the primary tumors. On the other hand, high EGFR expression levels in either primary tumors or metastatic tissues were associated with multiple metastatic disease. This suggests a negative prognostic role of EGFR expression. However, in a multivariate analysis, one-sidedness remained a strong independent predictive factor of survival. Previous studies demonstrated that the EGFR expression level depends on sidedness. Therefore, a subgroup analysis of the left- and right-sided cases was performed on both primary and metastatic tissues. In the case of metastic tissues, an analysis confirmed a better OS in low EGFR protein-expressing cases than in high EGFR protein-expressing cases. Collectively, these data suggest that EGFR protein expression is another negative predictive factor of the efficacy of cetuximab therapy of KRAS exon2 wild-type colorectal cancer.

## 1. Introduction

The predictive value of epidermal growth factor receptor (EGFR) protein expression has been debated since the introduction of anti-EGFR antibody therapies for the treatment of colorectal cancer [1] Even today, the protocol of cetuximab/Erbitux (but not of panitumumab/Vectibix) states that the drug can be used to treat EGFR-positive colorectal cancer [2]. Analysis of large clinical trials indicated that the efficacy of anti-EGFR antibody therapies of advanced colorectal cancer patients is independent from EGFR expression levels [3,4]. Earlier, there was even another report on the effectiveness of anti-EGFR agents in EGFR-negative cases [5]. These discrepancies could be due to inconsistencies in testing, evaluation differences and the use of various anti-EGFR antibodies [6,7]. Other studies suggest that activated EGFR could be a positive predictor of the efficacy of anti-EGFR therapy [8]. Using the breast cancer example of the close association between HER2 protein overexpression and HER2 amplification, several studies analyzed the connection between EGFR copy number variation (CNV) and protein expression. Unfortunately, the data concerning EGFR-CNV and the efficacy of anti-EGFR antibody therapy are conflicting. Earlier reports did not find any association between these two parameters [9], but more recently, amplification of EGFR was found to be a positive predictor of efficacy [10]. However, a clearcut association between EGFR-CNV and protein expression cannot be demonstrated [11]. On the other hand, mutations of the extracellular domain of EGFR (at codon 492 or 465) were found to be associated with cetuximab resistance [12,13]. Fortunately, strong negative predictors of the efficacy of anti-EGFR antibody therapies have been discovered: first KRAS exon2-, then rare KRAS- and NRAS mutations, and more recently BRAF mutations, the use of which improved patient selection for therapy [14,15,16]. Furthermore, it was also recently discovered that left-sided colorectal cancers respond significantly better to anti-EGFR therapies than do right-sided colorectal cancers [17,18]. Data also demonstrated that there is a difference in EGFR protein expression between left- and right-sided colorectal cancers that is independent from the RAS mutation statuses [11,19]. One explanation for this discrepancy is that these agents are used in metastatic diseases and EGFR protein expression of the primary tumor may not accurately reflect the status of the metastatic tissue. Meanwhile, the predictive role of EGFR protein expression in RAS wild-type colorectal cancer is still unknown. There is data indicating that EGFR protein expression in colorectal cancer is a negative prognostic factor associated with advanced-stage and lymphovascular invasion. It may also be associated with resistance to preoperative radiotherapy [20]. Another study demonstrated that the coexpression of EGFR protein and their ligands, TGFα, EGF, amphiregulin and betacellulin, could be a negative predictor of cetuximab effectiveness in metastatic colorectal cancer [21]. In a previous study, we have observed that RAS wild-type left-sided colorectal cancers are characterized by a lower EGFR protein expression as compared to those right-sided but respond significantly better to various anti-EGFR therapies [11]. This suggests that EGFR protein expression might be another negative predictive factor of anti-EGFR therapies. Therefore, we conducted a study on a homogenous cohort of 90 patients with KRAS exon2 wild-type metastatic colorectal cancer who had received cetuximab therapy. Long-term survival data were available for these patients.

## 2. Results

The study cohort was comprised of 90 KRAS exon2 wild-type metastatic colorectal carcinoma patients, all treated with the anti-EGFR antibody-(cetuximab)containing protocols. Primary tumors were available in 88 cases (77 resections and 11 biopsies). Only metastic tissue was obtained in the remaining two cases. Furthermore, in 27 of the cases, both primary tumors and their corresponding metastatic tissue were available for analysis. The localization of the primary tumor was predominantly rectosigmoideal (60/90). There was also a high prevalence of synchronous metastatic disease amongst the cohort (61/90). The predominant pT stage of the cohort was T3 (54/90, 60%) and the prevalence of the pN1-2 stage was 55/90 (61.1%). As expected, the majority of metastases occurred in the liver (55/90, 61.1%), followed by the lung (13/90, 14.4%), and then the two organs together (8/90, 8.9%) (Table 1).

EGFR protein expression of the primary tumors and the corresponding metastatic samples (all taken before the initiation of target therapy) was determined by immunohistochemistry (Figure 1). The level of EGFR protein expression at the tumor cell membrane was evaluated semiquantitatively using the H-score system (HS). In the case of multiple metastases, one sample/case was used. The median EGFR-HS was similar in both the primary and the metastatic tumor tissues (100 ± 66 versus 110 ± 75, respectively). The distribution of the EGFR-H-score levels (by 50 increments) was also very similar in the primary and the metastatic tumor tissues (Figure 2A). We systematically compared the HS of 27 metastases to their corresponding primaries and the individual alterations (decrease or increase) were plotted on Figure 2B. These data demonstrate that the metastases maintained the EGFR-HS range of the primary tumor only in a minority of cases (no difference, 3/27, 11.1%, ± 10% difference, 8/27, 29.6%). In the majority of cases, significant differences and extreme alterations in both directions (higher or lower) were found to occur in a random fashion (Figure 2B). We compared the EGFR H-score of the primary tumors with different metastatic potentials (i.e., single versus multiple metastatic diseases) and we found that EGFR protein expression is significantly higher in primary tumors with multiple metastases (*p* = 0.007, Figure 2C). Furthermore, comparison of the metastatic tissues of single versus multiple metastatic cases indicated that metastatic tissues of multiple metastases are characterized by a significantly higher EGFR-HS (*p* = 0.004, Figure 2C).

In this study, we analyzed the correlation between EGFR-HS and the progression-free survival (PFS) and overall survival (OS) of patients treated with cetuximab. We used Kaplan–Meyer statistics as well as widely different EGFR-HS threshold ranges (0, 50, 100, 200) to define low/high groups. Our data indicated that in the case of primary tumors with lower than threshold values, EGFR protein expression was associated with longer PFS and OS. However, the differences in relation to OS were statistically significant at the 200 threshold exclusively (*p* < 0.05) (Figure 3A,B). In the case of metastatic tissues, our data indicated that values lower than the applied threshold of EGFR-HS was associated with significantly longer PFS (at the 50 and 200 thresholds) and OS (at every threshold) and the difference was greatest at the lowest thresholds, gradually decreasing with the increasing EGFR-HS thresholds (Figure 3C,D, Appendix A).

Next, we applied the Cox proportional hazard model to test the predictive power of the EGFR H-score of primary tumors and metastases in a multivariate analysis using well-known factors such as sidedness, involved metastatic organs (single versus multiple), age and sex. This analysis indicated that EGFR H-score is a very weak independent predictor of OS in the case of cetuximab therapy. It approached significance only in the case of metastatic tissue, while sidedness is a much stronger predictor (Table 2 and Table 3). Age also turned out to be an independent predictor, but the corresponding relative risk (RR) levels were minimal. It is known that EGFR protein expression is significantly different in the case of left- versus right-sided cancers [11,19]. Therefore, we conducted a subgroup analysis on the OS of left- and right-sided cases based on EGFR-HS using Kaplan–Meyer statistics. EGFR-HS low- versus high status was determined by the median of the analysed subgroup. In the case of left- or right-sided primary tumors, there was no statistical difference in OS between EGFR-low and EGFR-high tumor cases (Figure 4A,B, respectively). However, a Kaplan–Meyer analysis based on EGFR expression of metastatic tumor tissues demonstrated that at both sides low EGFR-HS patients are characterized by a nominally better median OS (left side low: 766.5 days versus high: 368 days, right side low: 283.5 days versus high: 55 days). This finding was significant only in the case of left-sided tumors (*N* = 18, *p* = 0.016), likely due to the low number of right-sided metastatic cases (*N* = 11, Figure 4C,D, respectively).

## 3. Discussion

Our research provides evidence, for the first time, that low EGFR protein expression levels of tumor tissue are associated with significantly better survival of cetuximab-treated KRAS exon2 wild-type metastatic colorectal cancer patients. This result suggests that this marker could be another negative predictor of response. It is of note that the major differences in survival were found at the lowest EGFR-HS thresholds, 0 and 50, involving 9–27% of the treated KRAS exon2 wild-type patient population, respectively. We have also provided evidence for significant differences in EGFR protein expressions between primary and metastatic tumor tissue. This possibly explains the much better predictive performance of EGFR-HS detected in metastases. A multivariate analysis indicated that the EGFR protein expression of both the primary as well as the metastatic tissues is not an independent predictor of cetuximab efficacy unlike sidedness, confirming previous reports [17,18]. However, sidedness is not an EGFR-independent factor since it was shown recently that there is a close connection between EGFR protein expression and sidedness [11,19]: the poorly responding right-sided tumors express EGFR at significantly higher levels as compared to those left-sided [11]. Again, this suggests that EGFR protein expression is a negative predictive factor. Last but not least, analysis of the survival of left- and right-sided cetuximab-treated patients indicated, again, that higher than median EGFR protein expression is a negative predictor of efficacy on both sides. This was based on the EGFR data of the metastasis exclusively. On the other hand, we confirmed previous reports from the RAS mutation agnostic era (19) that a high EGFR protein expression of the KRAS exon2 wild-type primary colorectal cancer is also a negative prognosticator since expression levels were higher in colorectal cancers with multiple metastases when compared to those of single metastases.

Cetuximab recognises the ligand-binding extracellular domain of EGFR, while the Ventana antibody used here for immunohistochemistry [22] recognizes the juxtamembrane extracellular domain and therefore cannot detect any extracellular domain mutations or splice variations (vIII) observed in colorectal cancer [12,13]. Extracellular domain mutations and splice variations also negatively affect cetuximab efficacy. Therefore, further studies are needed to address this issue. However, the fact that the major differences in cetuximab efficacy were observed in the case of low EGFR protein-expressing tumors suggests that there may be other unknown mechanisms involved. Resistance to cetuximab therapy of KRAS exon2 wild-type colorectal cancer is under intensive study. One report revealed high EGFR and CCR7 protein overexpressions in tumors insensitive to cetuximab [23]. Another study found EPHA2 overexpression as a strong negative predictor for cetuximab efficacy [24]. Analysis of the tumors of the Prospect-C Cetuximab trial revealed miR-31-3p as a potential negative predictor of efficacy [25]. Last but not least, a systematic analysis of pre- and post-treatment tumors with acquired resistance to cetuximab therapy revealed a switch from consensus molecular signature-2 to consensus molecular signature-4 (a stromal expression signature), an increase in immune cell infiltrate and in the upregulation of checkpoint regulators. This suggests immune mechanisms of resistance [26]. Meanwhile, the issue of the protein target of cetuximab in colorectal cancer is still controversial and further research is much needed.

## 4. Matherials and Methods

### 4.1. Patients

We performed a retrospective analysis of 90 metastatic colorectal cancer patients who received cetuximab therapy at the Hungarian Defence Forces Health Center between 2008 and 2014. The analysis was approved by the local Institutional Review Board (IRB) (19/1043). All patients were diagnosed with multiple metastases. Sixty-seven patients had only one organ involvement, while 23 had multiple organ metastases. Synchronous metastatic disease characterized a majority of the cases (61/90), while metachronous metastasis development characterized the rest (29/90). In the majority of cases, the primary tumors were surgically resected (77/90). In the remaining cases, biopsies were only taken from the primary (11/90) or metastatic tumor tissues caused by synchronous metastases. Out of the 90 patients, 88 had biopsies or surgical samples from their primaries. The remaining two had biopsies only from their metastases. Twenty-seven metastatic samples from patients with evaluable primaries were available for comparison so a total of 29 metastatic tissues could be evaluated. All samples were taken before any anti-EGFR therapies. Out of the 29 metastatic samples, 11 were considered as metachron and 18 were synchronous metastases. Before anti-EGFR therapy, KRAS exon2 (75/90), and then extended K/NRAS exon2–4 (15/90) mutation analyses, were performed. The patients received cetuximab and FOLFIRI therapy with cetuximab administered as second-line therapy in the majority of the cases (63/90). Only three patients were alive at the end of the study. Clinical data of the patients are demonstrated in Table 1.

### 4.2. RAS Testing

Macrodissected FFPE primary tumor tissue was subjected to a KRAS exon2 mutation analysis with a technical sensitivity of 5% as described [27]. Due to protocol changes, extended RAS mutation analysis was performed using Idylla KRAS and NRAS mutation assays (Biocartis) in a smaller fraction of our cohort (15/90).

### 4.3. EGFR Protein Expression

The EGFR protein expression of primary and metastastatic colorectal cancer tissues was determined by immunohistochemistry using a Benchmark Ultra automatic stainer (Ventana, Tucson, AZ). In the case of multiple metastases, only one sample/case was used. EGFR protein was detected using a ready-to-use mouse monoclonal anti-EGFR antibody 3C6 (Confirm anti-EGFR antibody, Ventana). The specifically-bound antibody was revealed by an Ultraview Universal DAB detection kit (Ventana). Slides were counterstained with hematoxylin. The membranous EGFR protein expression level of tumors was determined by light microscopy using the industrial standard H-score semiquantitative methodology [22] (Figure 1) and was carried out in a blinded fashion by JT and IK. In the case of each tumor sample, 3 representative areas using a 40× lens were assessed for the percentage and intensity of EGFR-positive tumor cells. The intensity of the membrane labeling was defined in a 3-tier system (1 = weak, 2 = moderate, 3 = strong). In each intensity category, the %-age of the tumor cells was defined and multiplied with the respective intensity range (1–3). Finally, the H-score was calculated as the sum of the results of each intensity category. A completely negative result was defined as HS = 0 and a maximal HS was defined as 300 (100% of tumor cells with 3+ intensity).

### 4.4. Statistical Analysis

Based on the H-score of the EGFR, the studied patient cohort was divided into low and high expression groups according to the different cut-off values applied. Overall and progressive-free survival analyses were carried out using the Kaplan–Meier method. Survival intervals were determined as the time period from the start of the anti-EGFR therapy to the time of progression or death, respectively. The comparison between survival functions for different strata was assessed by log-rank statistics. A multivariate analysis was performed by the Cox proportional hazard model. In the case of numeric variables (age and EGFR H-score), risks were calculated by individual values. In the case of categorical variables (sex, sidedness, metastasis), subgroups were applied. Statistical significance was confirmed when *p* values were <0.05. A statistical analysis was performed using Statistica 13.0 software (StatSoft, Tulsa, OK).

## 5. Conclusions

This study strongly supports the notion that EGFR protein expression in colorectal cancer is a negative prognosticator. Furthermore, low- but not high-level EGFR protein expression on KRAS wild-type metastatic colorectal cancer cells may be a prerequisite for successful anti-EGFR antibody therapy. These paradoxical findings deserve further rigorous investigation.

## Figures and Tables

**Figure 1 cancers-12-00614-f001:**
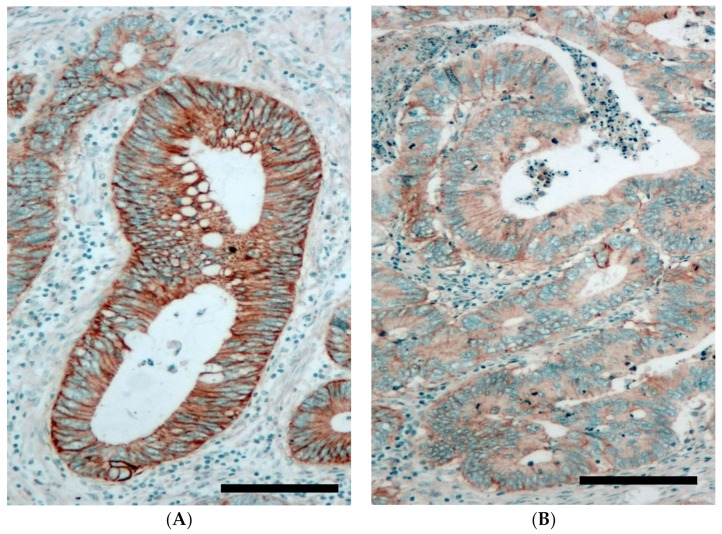
Epidermal growth factor receptor (EGFR) protein expression of colorectal cancer tissue as detected by immunohistochemistry (brown membrane signal). (**A**): High EGFR expressing primary colorectal cancer (H-score = 248). (**B**): Low EGFR expressing primary colorectal cancer (H-score= 31). Cell nuclei are stained by hematoxilin (blue). Bar = 200 µ.

**Figure 2 cancers-12-00614-f002:**
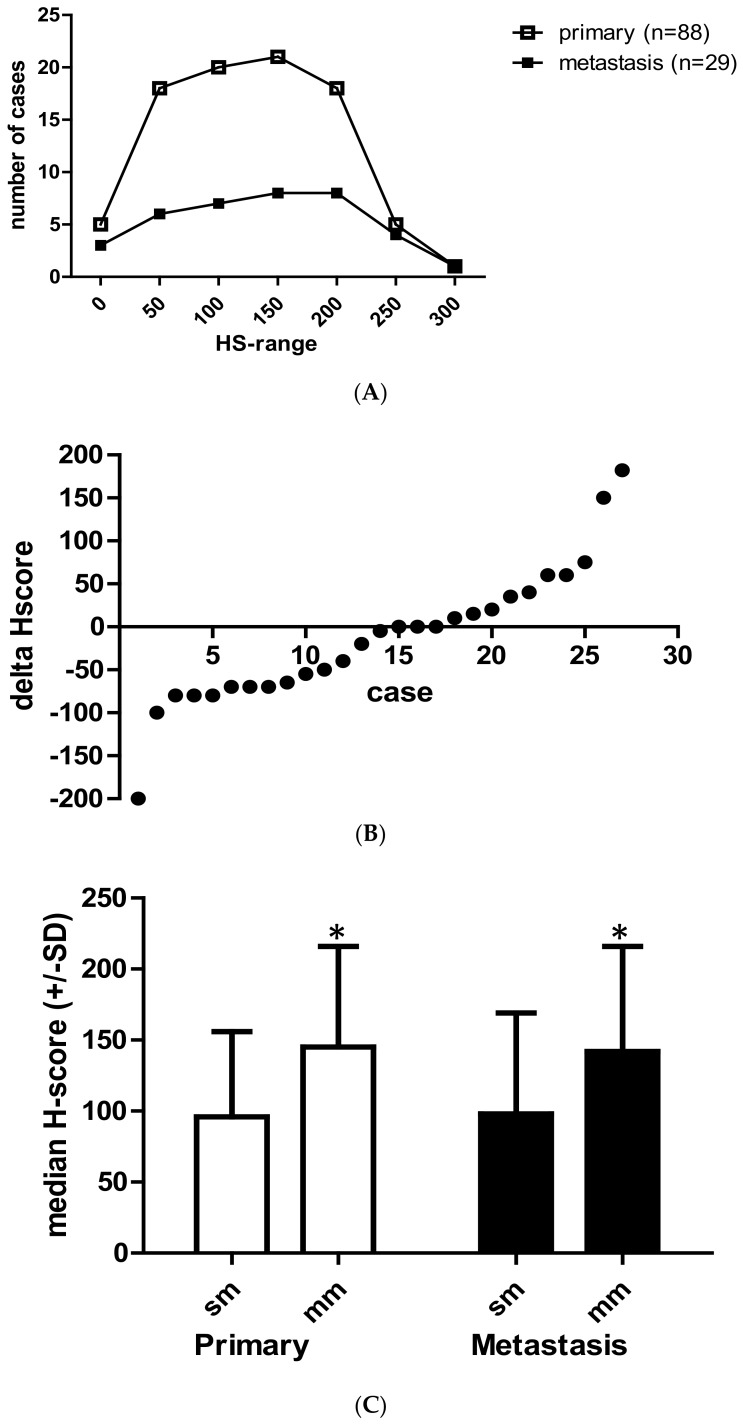
Comparison of the EGFR protein expression in primary and metastatic colorectal cancer tissues. (**A**) Distribution of EGFR expression levels in primary versus metastatic tumor tissues as represented with various H-score ranges. (**B**) Variations of EGFR-HS (H-score) in colorectal cancer metastases as compared to the corresponding primary tumor (*N* = 27). Data are expressed as H-score differences of metastatic minus primary tumor at individual case level. (0 = no change, negative value = decrease, + value = increase). (**C**) Comparison of EGFR H-score of the primary tumors with different metastatic potentials (single metastasis, sm, *N* = 22) versus multiple-metastasis, (mm *N* = 66), * *p* = 0.007. Comparison of metastatic tumors with single metastastasis (sm) versus multiple metastases (mm), ** *p* = 0.04. Data are expressed as median+/− SD, Mann–Whitney test.

**Figure 3 cancers-12-00614-f003:**
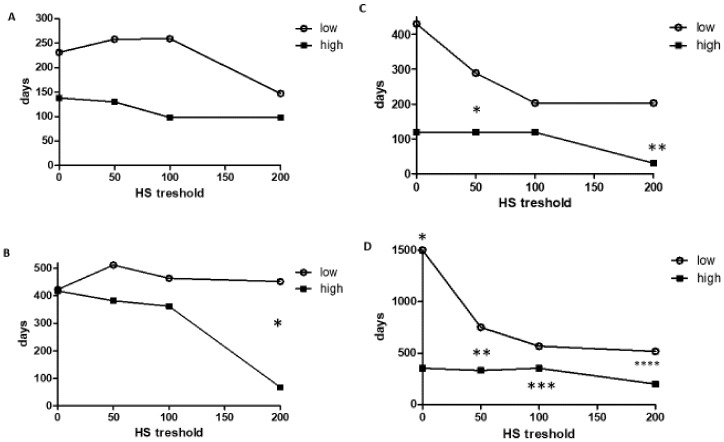
Progression-free and overall survival of cetuximab treated colorectal cancer patients (expressed in days) in relation to the level of EGFR protein expression defined as H-score. Kaplan–Meyer statistics. (**A**,**B**): Primary tumors (*N* = 88). (**C**,**D**): Metastatic tissue (*N* = 29). (**A**,**C**): Progression-free survival, (**B**,**D**): Overall survival in days. Low= below-, high=above HS threshold. (**B**): * *p* = 0.042, (**C**): * *p* = 0.024, ** *p* = 0.046, (**D**): * *p* = 0.008, ** *p* = 0.05, *** *p* < 0.018, **** *p* = 0.053.

**Figure 4 cancers-12-00614-f004:**
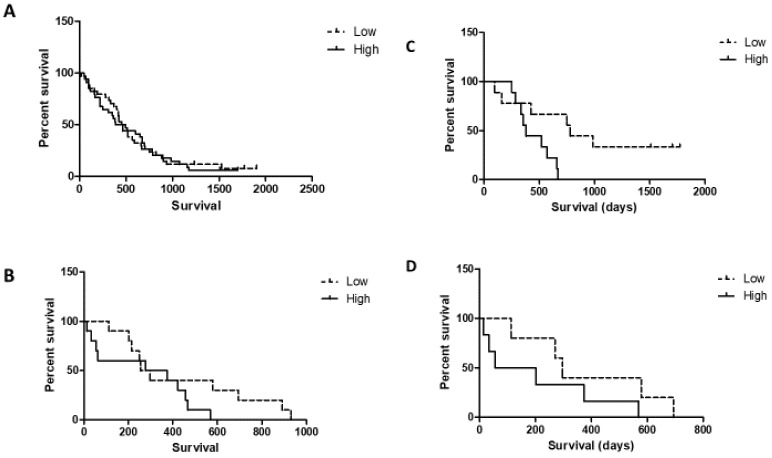
Overall survival analysis of left- and right-sided colorectal cancer patients treated with cetuximab stratified by the median EGFR-HS scores as low versus high of the primary tumors (*N* = 88) or metastastatic tissues (*N* = 29). (**A**) Left-sided primary tumors *N* = 69), EGFR HS median = 95. (**B**) Right-sided primary tumors (*N*= 19), EGFR HS median = 125. (**C**) Left-sided tumor metastases (*N* = 18), median = 80. (**D**) Right-sided tumor metastases *N* = 11), median = 150. Kaplan–Meyer statistics.

**Table 1 cancers-12-00614-t001:** Cetuximab-treated patient’s characteristics (*N* = 90).

Sex	[N]	[%]
Male	63	70.0
Female	27	30.0
**Age (years)**	[N]	[%]
Median	64	
Range	24–79	
**Primary Tumor Location**	[N]	[%]
Rectum	26	28.9
Rectosigmoid	6	6.7
Sigma	28	31.1
Descending colon	8	8.9
Lienal flexure	3	3.3
Transverse colon	4	4.4
Hepatic flexure	1	1.1
Ascending colon	8	8.9
Coecum	6	6.7
**Primary Tumor T**	[N]	[%]
NA	13	14.4
1	0	0.0
2	6	6.7
3	54	60.0
4	17	18.9
**Primary Tumor N**	[N]	[%]
NA	13	14.4
0	21	23.3
1	26	28.9
2	29	32.2
3	1	1.1
**Resection of Primary**	[N]	[%]
Yes	77	85.6
No	13	14.4
**RAS Testing**	[N]	[%]
KRAS exon2	75	83.3
Extended RAS	15	16.7
**Number of Evaluated Primary Tumors (*N* = 88)**	[N]	[%]
Right	19	21.6
Left	69	78.4
**Number of Metastases Evaluated by IHC (*N* = 29)**	[N]	[%]
Liver	17	58.6
Lung	2	6.9
Lymphnode	2	6.9
Cerebellum	1	3.4
Skin	1	3.4
Ovarium	1	3.4
Peritoneum	3	10.3
Soft tissue	1	3.4
Mesocolon	1	3.4
**Number of Metastases Evaluated by IHC (*N* = 29)**	[N]	[%]
Right	11	37.9
Left	18	62.1
Abbreviation(s): immunohistochemistry (IHC)		

**Table 2 cancers-12-00614-t002:** Multivariant analysis of various prognostic/predictive factors of cetuximab efficacy using Cox proportional hazard model of survival: EGFR protein expression of the primary tumors (*N* = 88).

Variables	*p*	RR (95% CI)
age	0.001	0.967 (0.943–0.993)
sex	0.715	1.099 (0.661–1.826)
EGFR-HS	0.53	1.001 (0.997–1.006)
sidedness	0.009	2.028 (1.193–3.448)
metastasis (S vs. M)	0.1	0.643 (0.38–1.089)

Multiple metastatic disease (M), relative risk (RR), single metastatic disease (S), sidedness = left or right.

**Table 3 cancers-12-00614-t003:** Multivariant analysis of various prognostic/predictive factors of cetuximab efficacy using the Cox proportional hazard model for overall survival: EGFR protein expression of the metastases (*N* = 29).

Variables	*p*	RR (95% CI)
age	0.001	0.916 (0.871–0.964)
sex	0.427	1.468 (0.57–3.784)
EGFR-HS	0.083	1.007 (0.999–1.016)
sidedness	0.006	5.694 (1.641–19.757)
metastasis (S vs M)	0.19	0.469 (0.151–1.455)

Multiple metastatic disease (MM), relative risk (RR), single metastatic disease (S), sidedness = metastasis derived from the left or right-sided primary.

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
