# Peer review of "EGFR Protein Expression in KRAS Wild-Type Metastatic Colorectal Cancer Is Another Negative Predictive Factor of the Cetuximab Therapy"

_cancers, 2020, doi:10.3390/cancers12030614_

Round 1
Reviewer 1 Report
The present study of Uhlyarik et al., provides informations that EGFR is a negative predictive factor in the therapy of patients with KRAS wild-type colorectal cancer with metastasis treated with the EGFR inhibitor cetuximab. A cohort of 90 patients with KRAS wild –type colorectal cancer treated with cetuximab was analyzed. 29 patients of this cohort were found to be positive for multiple metastasis. Via immunohistochemistry, the authors found higher EGFR protein expression in primary tumors with multiple metastasis. Kaplan-Meyer statistics indicated that lower expression of EGFR in primary tumors correlated with longer progression free or overall survival. From a multivariate analysis, the authors suggest that EGFR is not an independent predictor of cetuximab efficacy but sidedness and age. With regard to sidedness, right sided tumors expressing highly EGFR poorly respond to the treatment with cetuximab .
Overall, this study reveals evidences that the the expression levels of EGFR are an independent factor in the treatment of cetuximab in colorectal cancer patients with wild-type KRAS. However, this manuscript should be improved.
Major concerns
The authors should provide informations about the mutation status in their patient cohort. They can do this, in combination with table 1 or in a separate table. To reveal EGFR expression in their immunohistochemistry (IHC) of the colorectal cancer tissues, it is necessary that the authors present representative IHC-images. For the comparison of the left and right sided corectal cancers, the status of metastasis should be provided In Line 130, the authors state that “which was significant only in case of left-sided tumors (p=0.016), most probably due to the limited number of available right sided metastatic tissues”. My question here is, how many right sided metastatic tissues could be analyzed?
Minor concerns
The authors should increase the font size of all figures and the figures itself could be enlarged; Fig. 3, the authors presenting survival curves of left sided tumors and right sided tumors. For me, it seems that the outcome for patients with right sided tumors is better? – probably, since the authors used different scalings for the x-Axis. The authors should manage to present a standard for both curves. The content in line 151 – 154 is not clear and should be revised. The discussion and the form should be improved.Author Response
Please see the attachement

Reviewer 2 Report
The study proposed by Dr. Uhlyarik and colleagues is focused on demonstrating that low EGFR protein expression levels was associated with significantly better survival of the cetuximab-trated KRAS wild-type mCRC patients suggesting that EGFR protein expression could be a negative predictive factor.
The concept that is outlined is not new but could potentially provide clinically actionable results. However, is it still useful in post-RAS era?
English language and style are fine; however, in the introduction a more extensive overview of previous work would be advantageous.
Points to discussion:
Overall, the study seems to be technically sound however, the statement could be scaled down given the limited data at present (29 metastatic tissue samples).
Reporting median values instead of mean in Figure 1A would strengthen the author’s conclusions.
Only a subset of samples 25 out of 90 (1.6%) were also analyzed for NRAS and BRAF and no analyses for codon 61 and 146 of KRAS gene were performed. It is true that these mutations are rarer than the other analyzed (KRAS codon 12-13) however, they could be present in mCRC patients. A specific comment on this would be appreciated.
The authors compared the HS of 29 metastases to their corresponding primaries and the differences were plotted in Figure 1B. A minor problem consists in the number of plots reported, I can see 28 instead of 29 plots/samples. The major concern regards the data reported at page 3 line 84 that referred to all the 90 samples. How can the authors extrapolate these data if they have analyzed only 29 metastatic samples? Moreover, please make a comment on the differences and alterations in both direction (higher or lower) found. In my opinion, concerning the data on PFS and OS and the relative Figure 2, it is necessary to know how many samples were considered for every categories. I mean, how many samples are above or below the threshold 0, 50, 100 and 200? Are they enough to perform a statistical analysis? Please, provide this data introducing them in a table. Moreover, concerning the p-value reported in the caption of the same figure, they are not complete. In Figure D 4 star are shown but only a p<0.018 is referred. What about the other 3 stars? The same topic for Figure C.
The predictive power of EGFR score in multivariate analysis was performed on the 29 samples?
Is it right? If so, are they sufficient for this sub-classification?
Furthermore, how have the authors considered the parameter “age” that is reported to be significant?
Author Response
Please see the attachement

Reviewer 3 Report
Uhlyarik A. and colleagues investigated the expression of EGFR in a cohort of 90 KRAS exon2 wild-type metastatic colorectal carcinoma patients, all treated with anti-EGFR antibody- (cetuximab) containing protocols and suggested that the efficacy of cetuximab therapy is independent from EGFR protein expression of tumor cells. Although this observation is intriguing, and is in line with independent observations, the manuscript has considerable weaknesses and most of the results are inferred from a very limited number of cases, which suggests caution in drawing conclusions.
Below are specific points that should be addressed
Major points
The patient's cohort is not clearly described. Text (lane 70) indicates the presence of synchronous metastases in 61/90 patients, not reported in table 1, while 29 metachronous metastases are described in the table. I wonder if the 29 metastases included in this study refers to this group or to randomly selected synchronous metastases. This is a key point that compromises the understanding of all results. Figure 1a depicts the EGFR-HS levels in 90 primary tumors and 29 metastatic lesions. It is not clear whether they are synchronous metastases or are the same ones that are described in table 1. Lane 79-81: Based on the EGFR-HS levels, three groups have been identified: 0,1-100; 101-200; 201-300. However, primary and metastatic tumor tissues are subdivided into four groups. Specifically, 5.2%, 46.4%, 41.2% and 2.7% (primary) versus 10%, 48.5, 33.3,10 (metastases). What does it mean? Figure 2. The interpretation of these data requires caution given the small number of samples analyzed, especially in panels C and D. No definitive conclusions can be drawn (see Discussion, lane 167-168 and Conclusion). What would be the results if patients were dicotomised into high and low based on the median of EGFR-HS? Overall survival analysis of left and right-sided colorectal cancer patients stratified by the median of EGFR-HS of the primary tumor must be shown. The impact of the same analysis performed on metastatic tissues is weak due to the small number of cases analysed. EGFR expression was determined with an antibody that recognizes an epitope other than cetuximab, and it is unable to discriminate different variants of the receptor, including variants that do not bind cetuximab. In order to conclude that receptor expression is a negative predictor of cetuximab efficacy, it is mandatory to evaluate the EGFR expression in each tissue sample with cetuximab.Minor points
Figure 1B. It is not clear if each point indicates the difference between EGFR-HS in primary tumor and corresponding metastasis (lane 82) or it indicates the ratio between the two (legend to figure 1). Figure 1C. How many single metastatic lesions and multiple metastatic lesions are depicted? How was the HS of multiple metastases calculated? Figure 1. Y axis in panel A and B are indicated in a different way. Significance is missing in panel C. Figure 3. The numbers of patients at risk on day 0, 500, 1000, 1500, 2000 should be indicated below the graph.
Author Response
Please see the attachement

Reviewer 4 Report
This manuscript describes that EGFR protein expression was measured using tumor tissue samples and evaluated using H-score methodology. The authors suggested that EGFR protein expression is a negative predictive factor for cetuximab therapy. Overall, the manuscript is convincing in its comprehensive methodical approach. Although I am left with the impression that there are a couple of pieces to the puzzle remaining to be discovered in this model, there is sufficient information in this manuscript to warrant publication at this time.
Comments:
Please add a precise description about the EGFR-HS score. For example, how is it evaluated, and who measured this? Please add a description of the ligands of EGFR and EGFR expression and their association with cetuximab efficacy. Please add the mark of statistical significance in Figure 1. Page 5, Line 113: Please add a description of the meaning of “selected.” Page 5, Line 127, and Page 8, Line 209: Please add the meanings of the expressions “low” and “high” precisely. Page 7, Line 158: What is indicated by “median” in “median EGFR protein expression”? Page 7, Line 174: Please define the abbreviations CMS2 and CMS4.Author Response
Please see the attachement

Round 2
Reviewer 1 Report
Comments to the authors
Throughout the manuscript, there are still a lot of errors (including the form), which should be revised by the authors:
- Line 27, a space character is lost between KRAS and exon2. This mistake appears throughout the text;
- Line 125, the authors wrote “whith”, not with;
- Line 177, after (Table), a dot is missing;
- In the discussion, the authors refer to the reference (19), which can not be followed up, since it is not included in the reference list. To this, the reference number (12) appears two times in the reference list, but two different citations refer to this number;
- In line 227, the authors using the term “metastases”. Do the authors mean “distant metasteses”? If not, they should use the right form, “metastasis”;
- In line 228, what is the meaning behind “pre-RAS era”? Never heard about it, before! Also, this expression has been linked to the reference number (19), which is not in the reference list;
- In line 243, the authors refer to “the NewEPOC Cetuximab trial”, in relation to reference number (25); obviously, the authors did not read or understand the corresponding reference (25) exactly, since this paper is about the “PROSPECT-C trial (ClinicalTrials.gov identifier: NCT02994888);
- In Figure 1, the authors should add a scale bar to the images;
- Figure 2A, the labeling of the x-axis is not given and also the legend for the primary tumors is missing;
- Figure 2C, in the representative figure legend, the authors using the abbreviations (M) and (P), for what reason?
- Table 2+3, the authors using commas instead of dots. The authors should keep to the international English standard and not using the European;
Author Response
Responses to Round 2. Reviewer 1.
Throughout the manuscript, there are still a lot of errors (including the form), which should be revised by the authors:
- Line 27, a space character is lost between KRAS and exon2. This mistake appears throughout the text;
Please find the corrections in the latest uploaded version with yellow.
- Line 125, the authors wrote “whith”, not with;
Please find the correction in the latest uploaded version with yellow.
- Line 177, after (Table), a dot is missing;
Please find the correction in the latest uploaded version with yellow.
- In the discussion, the authors refer to the reference (19), which can not be followed up, since it is not included in the reference list. To this, the reference number (12) appears two times in the reference list, but two different citations refer to this number;
Sorry for the mistake, the reference list numbering was corrected, please see the latest version, corrected numbers in yellow..(ref19-21) see line264 and line422-424
- In line 227, the authors using the term “metastases”. Do the authors mean “distant metasteses”? If not, they should use the right form, “metastasis”;
Please find the correction in the latest uploaded version with yellow. line 263
- In line 228, what is the meaning behind “pre-RAS era”? Never heard about it, before! Also, this expression has been linked to the reference number (19), which is not in the reference list;
In line 264 we attempted to clarify this issue, changed the expression of„ pre-RAS era” to
RAS-mutation agnostic era.
- In line 243, the authors refer to “the NewEPOC Cetuximab trial”, in relation to reference number (25); obviously, the authors did not read or understand the corresponding reference (25) exactly, since this paper is about the “PROSPECT-C trial (ClinicalTrials.gov identifier: NCT02994888);
Sorry for the mistake, please find the corrected title, PROSPECT-C, in line of 279 of the final uploaded version.
- In Figure 1, the authors should add a scale bar to the images;
We adedd the requested scale bar to Figure 1. in the latest uploaded version 3.
- Figure 2A, the labeling of the x-axis is not given and also the legend for the primary tumors is missing;
We have removed the Figure 2A as other reviewrs considered it unnecessary, and included the relevant information into the text of Results, please see line 121-134.
- Figure 2C, in the representative figure legend, the authors using the abbreviations (M) and (P), for what reason?
The abbreviations (M, P) have been deleted.
- Table 2+3, the authors using commas instead of dots. The authors should keep to the international English standard and not using the European;
Please find the corrections in the latest uploaded version .
Reviewer 2 Report
I appreciated the effort of the author to respond my observations and criticisms performing the required changes.
However, I will remark some points in an analysis below.
- It is difficult for me understand how the median of H-score in Fig 2A could be so similar in all primary and metastatic tumors available if when we compare cancer metastasis to the corresponding primary tumor the individual alteration are so evident (Fig 2C). It is due to the different sample size? In my opinion, Fig 2A in useless and confusing.
- Table 1 still remain quite chaotic. First of all, I suggest to report only the “information” concerning the 88 samples analyzed rather than the 90 enrolled. The same for the 27 samples in which were available both primary and metastatic samples (rather than 29).
Moreover, I suggest moving the “RAS” testing above the “metastases evaluations”.
- Concerning my previous point number 5.
Response of the authors
These data (new Fig. 2C) demonstrate that in this subset of the cohort the metastases maintained the EGFR-HS range of the primary tumor only in a minority of cases extrapolated from the data of the 27 pairs (10% variations in 10/90 cases, 20% variation, 17/90 cases), whereas in the majority of cases significant differences and extreme alterations in both directions (higher or lower) were found to occur statistically in a random fashion. Since we haven’t got access to more cases with primary and corresponding metastatic tissues we used that subcohort. We are not aware of any study previously which ever performed such a comparison of EGFR-HS of metastasis compared to primary.
I perfectly understood what the Fig.2C represented. However, in my opinion it is not correct refer to all the 90 samples if only 27 samples were analyzed. I suggest to report the percentage of variation using n/27.
- Please, make a comment on the results that age is an independent predictor.
Author Response
Response to Round 2. Rew.2.
I appreciated the effort of the author to respond my observations and criticisms performing the required changes.
However, I will remark some points in an analysis below.
- It is difficult for me understand how the median of H-score in Fig 2A could be so similar in all primary and metastatic tumors available if when we compare cancer metastasis to the corresponding primary tumor the individual alteration are so evident (Fig 2C). It is due to the different sample size? In my opinion, Fig 2A in useless and confusing.
Response: We accepted the critique and omitted the Fig2A and provide data in Result text only (line 119)
- Table 1 still remain quite chaotic. First of all, I suggest to report only the “information” concerning the 88 samples analyzed rather than the 90 enrolled. The same for the 27 samples in which were available both primary and metastatic samples (rather than 29).
Response: All of the 90 enrolled patient were treated with cetuximab .Therefore we collected data of all cases. From the 90 patient in 88 cases samples from the primaries ( either surgical or biopsy) were available, but in two synchron metastatic cases only metastatic tissue were taken in order to prove the diagnosis. We attempted to clarify this issue in line 299-302: "From the 90 patient 88 had biopsy or surgical samples from their primaries, while 2 had biopsy only from their metastases. This way 27 metastatic samples from patients with evaluable primaries were available for comparison, all together 29 metastatic tissues were evaluated."
Moreover, I suggest moving the “RAS” testing above the “metastases evaluations”.
Response: we have corrected the Table 1. as requested.
- Concerning my previous point number 5.
Response of the authors
These data (new Fig. 2C) demonstrate that in this subset of the cohort the metastases maintained the EGFR-HS range of the primary tumor only in a minority of cases extrapolated from the data of the 27 pairs (10% variations in 10/90 cases, 20% variation, 17/90 cases), whereas in the majority of cases significant differences and extreme alterations in both directions (higher or lower) were found to occur statistically in a random fashion. Since we haven’t got access to more cases with primary and corresponding metastatic tissues we used that subcohort. We are not aware of any study previously which ever performed such a comparison of EGFR-HS of metastasis compared to primary.
I perfectly understood what the Fig.2C represented. However, in my opinion it is not correct refer to all the 90 samples if only 27 samples were analyzed. I suggest to report the percentage of variation using n/27.
Response: We have corrected the sentence concerning fig 2newB. see Line 125 providing % for n=27 metastatic cases exclusively.
- Please, make a comment on the results that age is an independent predictor.
Response: the multivariate analysis demonstrated that age is statistically significant independent predictor in case of both the primary and metastatic tissues-based analysis however the RR affecting survival is minimal. This comment can be seen in line 212,213.
Reviewer 3 Report
The authors made a considerable effort to improve their manuscript, which in its present form can be accepted for publication.
Author Response
Thank you for your previous response.
"The authors made a considerable effort to improve their manuscript, which in its present form can be accepted for publication."
Round 3
Reviewer 2 Report
I appreciate the effort of the authors to replicate my clarifications.
Author Response
We would like to say thank you for your support.